# Cable bacteria reduce methane emissions from rice-vegetated soils

Vincent V. Scholz [1✉], Rainer U. Meckenstock[2], Lars Peter Nielsen [1] & Nils Risgaard-Petersen[1]

Methane is the second most important greenhouse gas after carbon dioxide and approximately 11% of the global anthropogenic methane emissions originate from rice fields. Sulfate amendment is a mitigation strategy to reduce methane emissions from rice fields because sulfate reducers and methanogens compete for the same substrates. Cable bacteria are filamentous bacteria known to increase sulfate levels via electrogenic sulfide oxidation. Here we show that one-time inoculation of rice-vegetated soil pots with cable bacteria increases the sulfate inventory 5-fold, which leads to the reduction of methane emissions by 93%, compared to control pots lacking cable bacteria. Promoting cable bacteria in rice fields by enrichment or sensible management may thus become a strategy to reduce anthropogenic methane emissions.

[1] Center for Electromicrobiology, Aarhus University, 8000 Aarhus C, Denmark. [2] Environmental Microbiology and Biotechnology, University Duisburg-Essen, 45141 Essen, Germany. ✉email: vincent.scholz@bios.au.dk

Cable bacteria are filamentous bacteria of the family Desulfobulbaceae[1], which spatially decouple the oxidation of sulfide and the reduction of oxygen[2] or nitrate[3] in marine[4,5] and freshwater systems[6,7] by channeling electrons along their filaments. This process is called electrogenic sulfide oxidation (e-SOX) and results in the centimeters-wide depletion of sulfide and accumulation of the end-product sulfate[8–10]. The sulfate inventory is further fueled by the dissolution of iron sulfides through the release of protons from e-SOX and the subsequent oxidation of the liberated sulfides to sulfate[10,11]. Moreover, the establishment of an electric field through e-SOX causes downward migration of sulfate and minimizes its loss to the water column[6]. Hence, the electric field contributes to the e-SOX-driven accumulation of sulfate, which has recently been shown to stimulate sulfate reduction[12].

External amendment of sulfate-containing compounds such as ammonium sulfate or gypsum to wetland rice soils is known to mitigate methane ($CH_4$) emissions[13]. This is because additions of sulfate stimulate sulfate reduction in otherwise sulfate-limited environments, which leads to substrate limitation of methanogenesis, as sulfate-reducing microorganisms are superior to methanogens as competitors for common substrates such as hydrogen and acetate[14,15].

However, the externally amended sulfate can be quickly converted to sulfide, which tend to accumulate in the soil, implying that prolonged effects on $CH_4$ emissions can only be achieved by re-application of low amounts of sulfate[16,17]. Here, we carried out rice pot experiments with autoclaved wetland soil which was mixed with cow dung to inoculate a complex microbial community lacking cable bacteria. Half of the pots were then inoculated with the freshwater cable bacteria Ca. Electronema sp. GS[9]. After 11 weeks of incubation under water saturation, sulfate concentrations, $CH_4$ emissions, and cable bacteria abundance and activity were determined. Our results indicate that cable bacteria reduce $CH_4$ emissions from rice-vegetated soils by recycling sulfate via e-SOX.

## Results

**Cable bacteria distribution**. After the 11-week incubation period, no difference of the above-ground biomass of the rice plants between the treatments was observed ($P = 0.64$, $n = 8$, unpaired two-tailed $t$-test, see Supplementary Fig. 1). Fluorescence in situ hybridization (FISH) showed $400 \pm 100$ m cm$^{-2}$ cable bacteria filaments in the inoculated pots, demonstrating that the cable bacteria could be successfully transferred to cable bacteria-free soils and grew to high densities. Most of the cable bacterial cells were located in the uppermost 2 cm (Fig. 1a, Supplementary Fig. 2a). Moreover, cable bacteria were found in close contact with the rice roots (Fig. 1b, Supplementary Fig. 2b). Nevertheless, the observed higher filament density in the upper soil layers

indicated that most of the cable bacteria reduced oxygen diffusing from the water column into the soil. No cable bacteria filaments were seen in the control rice pots after the 11-week incubation period.

**Sulfate and pH depth profiles**. The sulfate concentration in the overlaying water phase of the cable bacteria-free and cable bacteria incubation was $1646 \pm 3$ and $2230 \pm 20$ μM ($n = 3$, technical replicates), respectively. Without cable bacteria, the sulfate concentration of the soil porewater declined from $1000 \pm 100$ to $70 \pm 20$ μM in 4 cm depth, indicating that the sulfate diffusing from the water column into the soil was reduced with no reformation by sulfide oxidation in deeper soil layers (Fig. 2a). With cable bacteria, the sulfate inventory of the soil porewater in the upper 4 cm ($90 \pm 20$ mmol sulfate m$^{-2}$) was five times higher ($P = 0.006$, $n = 8$, unpaired two-tailed $t$-test) than in pots without cable bacteria ($17 \pm 2$ mmol sulfate m$^{-2}$) and the sulfate concentration was uniform throughout the first 4 cm of the soil, ranging from $1900 \pm 100$ μM in the top centimeter to $2200 \pm 700$ μM at 3–4 cm depth (Fig. 2a). The experimental design did not allow to retrieve soil samples from the lower 3 cm of the pots, but the facts that the sulfate concentrations in the 3–4 cm depth section of the pots with cable bacteria were about twice as high as the concentrations in the top centimeter of pots without cable bacteria and sulfate yet penetrated 4 cm in the latter do imply that sulfate penetrated to the bottom of the cable bacteria pots, assuming similar potential sulfate reduction rates.

Furthermore, typical effects of e-SOX[6,10] in the cable bacteria-amended pots developed during the incubation period. The pH decreased by 0.24 units in 7.2 mm depth (Fig. 2b) and an orange layer formed on top of the soil surface (Supplementary Fig. 1), which probably originates from dissolution of iron sulfides and subsequent diffusion, oxidation, and precipitation of iron as ferric iron oxides[18]. The sulfide from dissolution of iron sulfide gets oxidized to sulfate by e-SOX and most likely contributed to the overall sulfate inventory in pots with cable bacteria.

**$CH_4$ emission**. $CH_4$ emission rates from the pots were calculated from the linear increase of $CH_4$ in the incubation system (Fig. 3). The $CH_4$ emission from the pots with cable bacteria was significantly ($P = 0.006$, $n = 8$, unpaired two-tailed $t$-test) lower than the emission from the pots without cable bacteria ($42 \pm 9$ vs. $600 \pm 100$ μmol m$^{-2}$ day$^{-1}$)). Thus, presence of cable bacteria led to a reduction of $CH_4$ emissions by 93%.

## Discussion

The cable bacteria-mediated 93% reduction of $CH_4$ emission is one of the highest reported reduction efficiencies compared to studies where sulfate was added[13,16,17]. The main controlling

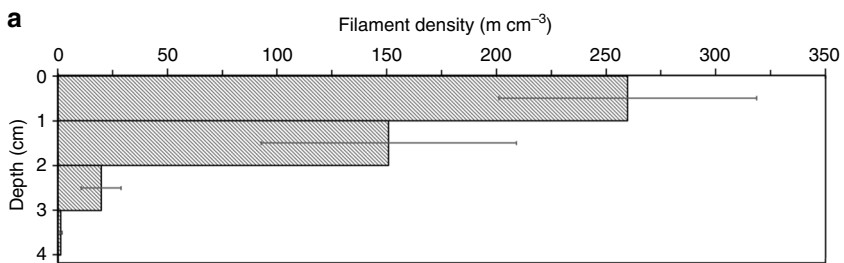
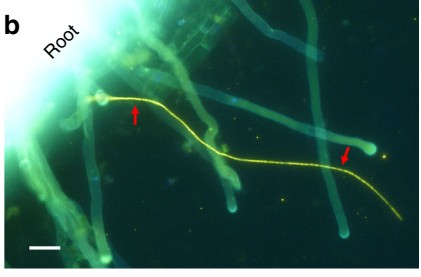

**Fig. 1 Distribution of cable bacteria in rice-vegetated pots. a** Depth profile of the cable bacteria filament densities in inoculated rice soils. Data are presented as mean ± standard error of the mean ($n = 4$). **b** Micrograph of a cable bacteria filament in close contact with a rice root. Image from DAPI staining (blue) was superimposed with FISH images hybridized with probe DSB706 specific for Desulfobulbaceae labeled with Cy3 (red) and probe EUB-MIX targeting most bacteria labeled with Atto-488 (green). Red arrows point to the cable bacteria filament. Scale bar, 20 μm.

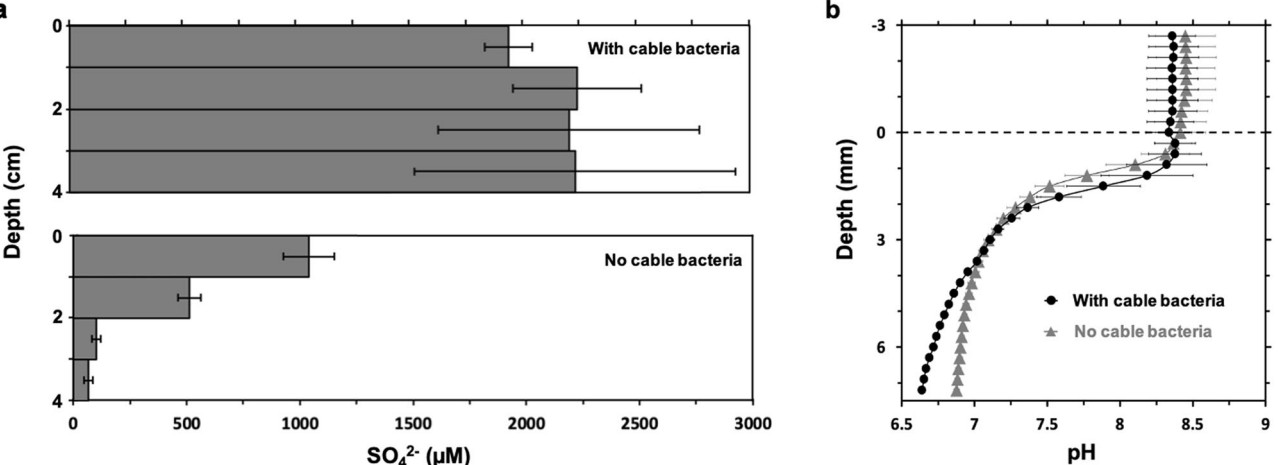

**Fig. 2 Sulfate concentrations and pH in the rice-vegetated pots. a** Sulfate concentrations in the porewater of pots with cable bacteria (top) and in pots without cable bacteria (bottom). **b** Depth profiles of pH measured with microelectrodes in pots with cable bacteria (circles) and without cable bacteria (triangles). Data are presented as mean ± standard error of the mean ($n = 4$).

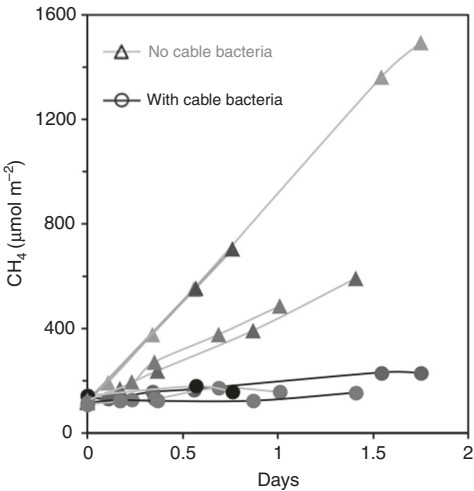

**Fig. 3 CH₄ emissions.** The emitted amount of $CH_4$ from replicate rice pots with cable bacteria (circles) and without cable bacteria (triangles) normalized to the surface area of the pots as a function of time.

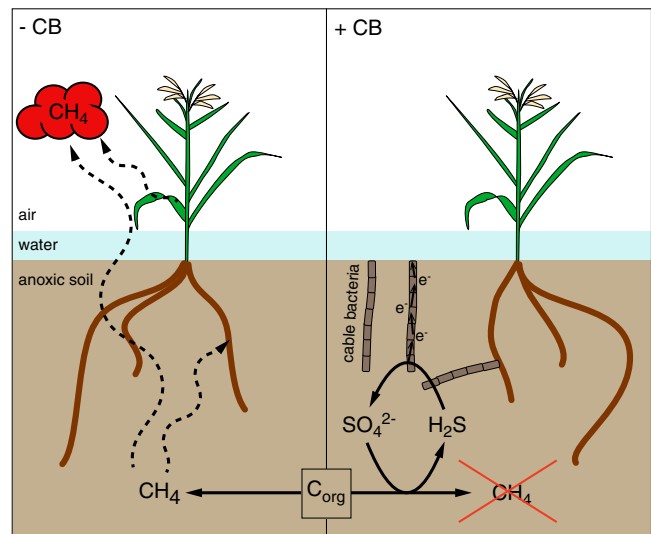

**Fig. 4 Potential microbial-mediated processes.** In rice-vegetated soils without cable bacteria (left) and with cable bacteria (right) organic carbon ($C_{org}$), e.g. from root exudation, is degraded by a consortium of microorganisms to acetate and hydrogen, which are the main substrates for methanogens. In soils with cable bacteria, increased sulfate concentrations due to e-SOX stimulate sulfate-reducing bacteria, which outcompete methanogens for common substrates. The produced $CH_4$ can enter to the atmosphere through the soil and the plants (dotted arrows).

factor in our experiments was likely the sulfate accumulation which was uniformly distributed in the upper 4 cm in pots with cable bacteria, suggesting that sulfate reduction was balanced by sulfur re-oxidation via e-SOX and eventually that ionic migration adds to the transport of sulfate[12]. The sulfate accumulation stimulated the activity of sulfate-reducing bacteria and therefore outcompeted methanogens for common substrates such as hydrogen or acetate. These substrates were supplied from fermentations processes, which were fueled by organic carbon from root exudations and sloughed root material[19]. Moreover, we chose autoclaved wetland soil as universal matrix to test our hypothesis. This wetland soil was supplemented with cow dung to provide an inoculum consisting of a complex microbial community including fermenters, methanogens, and sulfate-reducing bacteria[20] but no cable bacteria, which also increased the organic carbon pool in the incubation pots at the beginning of our experiment. Beyond augmenting sulfate, e-SOX produced acidity. Microprofiles of pH were only taken down to 7.2 mm depth to minimize the risk of sensor collision with the roots which would break the sensor. However, it has been shown previously that e-SOX produces acidity even down to 4 cm[12]. Low pH values

impede fermentation processes and methanogenesis, resulting in the reduction of methane emissions[21,22]. For example, a pH shift from pH 6.9 to 6.8 has been reported to decrease $CH_4$ production in flooded rice soil by 26%[23]. Taken together, the multifarious effects of e-SOX, i.e. sulfate accumulation and pH decrease, may explain the strong impact on $CH_4$ emissions (Fig. 4).

In contrast to sulfate amendments as mitigation strategy where the effect on $CH_4$ emissions weakens after the application[17], the cable bacteria-mediated sulfate accumulation was actively maintained through e-SOX even after 11 weeks incubation. Furthermore, external amendments of sulfate may result in the build-up of toxic sulfide concentrations[17,24] whereas e-SOX prevents such build-up[2,6,10] and possibly even promotes the plant performance through provision of sulfate as nutrient[25].

Cable bacteria successfully established after one-time inoculation and the filament density was well within the typical range of cable bacteria abundance[6,26]. Cable bacteria were also found on roots which is congruent with previous studies that report the enrichment of cable bacteria on oxygen-releasing plant roots[27,28]. Indeed, rice roots can release oxygen[29] providing the electron acceptor for cable bacteria. Thus, wetland rice fields might constitute an ideal habitat for cable bacteria. Our finding arises the questions to what extent cable bacteria grow in wetland rice fields and whether their presence can be promoted by the one-time inoculation with cable bacteria at the beginning of the rice cultivation period and through adjusted water levels to keep the top soil layer well oxygenated throughout the cultivation period.

## Methods

**Plant growth.** Seedlings of *Oryza sativa* were germinated in tap water and grown to the three-leaf stage in commercially purchased garden soil under drained conditions. Wetland soil was retrieved from a small eutrophic lake in Aarhus, Denmark (56°09′53.32N, 10°12′28.73E) in April 2019, sieved, and autoclaved. Freshly collected cow dung was mixed with the autoclaved soil. The soil was then inoculated with cable bacteria by adding cable bacteria enrichment culture of *Ca.* Electronema sp. GS[9] to the soil–cow dung mixture (≈1500:1, v/v). After carefully homogenizing the soil and filling into growth pots (depth: 7 cm; diameter: 12 cm), the first 2 cm of the soil in each pot were further inoculated with the cable bacteria enrichment culture (≈225:1, v/v) and carefully homogenized. After transplanting the rice plants, the four replicates of each treatment were placed into incubation tanks with aerated tap water. Cross-contamination was excluded by avoiding direct contact between the two incubation tanks. The water level was continuously kept above the soil surface up to several centimeters. The plants were grown at room temperature and at the window under natural light conditions for 11 weeks with additional illumination in the first weeks. After sampling, the above-ground biomass from each rice plant was collected and dried at 80 °C for 44 h to determine the dry weight.

**Microelectrode measurements.** After the 11-week incubation period, depth profiles of pH were recorded with custom-made microsensors[30] and a commercially available reference electrode (Red Rod reference electrode, REF201, Radiometer Analytical, Denmark). The software SensorTrace Pro (Unisense A/S, Denmark) was used to operate the micromanipulator and for data acquisition. The pH sensors were calibrated in buffers of pH 4, 7, and 10 (HANNA instruments, UK) and the depth of each profile was corrected in MS Excel (Microsoft Corporation, United States).

**CH₄ measurement.** Following the microelectrode measurements, the pots were taken out of the water tank and any remaining water on top of the soil was carefully removed. Pots were placed into a custom-made opaque PVC chamber (inner dimensions: 91 cm height, 14 cm diameter) with a rubber septum at the top and incubated up to 2.5 days. Headspace samples of 500 μL were withdrawn with a syringe and directly injected into a gas chromatograph (310 C, SRI Instruments, United States) equipped with a flame ionization detector.

**Sulfate and FISH analysis.** One soil core with an inner diameter of 4.5 cm from each pot was taken and sliced into four sections of 1 cm width. For sulfate measurement, the porewater from each depth section was separated from the solid phase by centrifugation, filtered through 0.22 μm, and stored at 6 °C until analysis by ion chromatography (Dionex, USA) with a AG18 Guard column and Dionex IonPac AS18 column (Thermo Fisher Scientific, USA). The run time was 18 min with sulfate eluting at 9.5 min.

For FISH analysis, 0.5 mL soil from each section was mixed with 0.5 mL ethanol and stored at −20 °C. After taking the core, roots sections were cut out of the remaining soil with scissors, stored, and washed in 50% ethanol, and dried and embedded in 0.5% agarose on a well-slide. Cable bacteria filaments were stained and quantified by FISH as described earlier[27].

**Statistical analysis.** Results are displayed as mean ± s.e.m. of four biological replicates unless stated differently in the text. The dry weight of the above-ground biomass, CH₄ emissions rates, and depth integrated sulfate concentrations were tested for difference between rice pots with and without cable bacteria using the unpaired two-tailed Student's *t*-test with the significance level of 0.05, eight observations and six degrees of freedom.

The experimental design is illustrated in Supplementary Fig. 3.

**Reporting summary.** Further information on research design is available in the Nature Research Reporting Summary linked to this article.

## Data availability

The raw data generated in this study are available from the corresponding author upon request.

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

## Acknowledgements

We acknowledge Ronny Mario Baaske, Mette L.G. Nikolajsen, and Theresa Merl for assistance in rice cultivation; Lars Borregaard Pedersen for making the microsensors; and Karina Bomholt Oest for help with the sulfate measurements. Casper Thorup and Ronny Mario Baaske are thanked for providing sediment enrichments of *Ca.* Electronema sp. GS. Korneel Rabaey is thanked for help in conceiving the hypothesis. This research was financially supported by the Danish National Research Foundation DNRF136.

## Author contributions

V.V.S. designed and carried out the experiment and wrote the manuscript with input from all the co-authors. R.U.M. conceived the idea of the project and contributed to the design of the experiment. L.P.N. helped with interpretations of the results. N.R.-P. guided the research and assisted with data analysis.

## Competing interests

The authors declare no competing interests.
