## [Peer Review File · Nature Communications]

Reviewers' comments:

Reviewer #1 (Remarks to the Author):

In this manuscript, the authors show that inoculation of rice plant-vegetated soil pots with filamentous cable bacteria reduced methane emissions by 92%. They showed that sulfate concentrations increased significantly in pots amended with cable bacteria. This sulfate could then serve as an electron acceptor for sulfate reducers that were able to outcompete methanogens.

Nice story. I enjoyed reading this paper. I just have a few minor comments but otherwise I think that this is an excellent paper.

Line 34. 11-week incubation period, cable....

Lines 63-66. This sentence is awkward, please reword

Line 66. We hypothesized that cable bacteria could reduce methane emissions.....

Line 71. After 11 weeks of incubation...

Line 73. bacterial cells

Line 100. What do you mean by domain? Are you referring to this specific region (upper 4 cm)? Please clarify.

Line 106. Did it penetrate deeper? You shouldn't have to speculate. Why didn't you just measure sulfate concentrations below 4 cm?

Line 106-107. I'm not sure what you are getting at here. Please rewrite this section. Are you just trying to say that sulfate reducers are only present in the upper layers of the pots (where sulfate was found in the cable-free pots)???

Lines 136-137. But you didn't see drops in pH until after 4 cm of depth (Figure 2b). You didn't show data below 4 cm. What were sulfate concentrations like at depths where low pH would influence methanogenesis? Just curious.

Line 152. Provide a reference showing that sulfate might improve plant performance by serving as a nutrient.

Line 168. mixed with the autoclaved soil to provide an inoculum consisting of a complex microbial community

Line 170. Please be clearer about the number of tablespoons that were added. Do you mean three tablespoons were added? Did you add exactly the same amount of cable bacteria to each amended pot?

Reviewer #2 (Remarks to the Author):

The manuscript, "Cable bacteria reduce methane emissions from rice-vegetated soils" details an experiment in which rice seedlings were planted in pots with a mix of wetland soil and cow dung (as microbial inoculum) with or without an inoculum of cultured freshwater filamentous cable bacteria (*Ca. Electronema* sp. GS). With the addition of the cable bacteria, the authors show sulfate was recycled, porewater pH was lowered, and methane emissions were mitigated to a remarkable degree. The authors suggest that the decrease in methane emissions were due to the combined effects of sulfate

regeneration and acidity generation, as a result of cable bacteria activity near the sediment surface. Recently, Scholz et al. 2019 (FEMS) reported observing cable bacteria in the rhizosphere of several plants, including *Oryza sativa* (rice), though at far lower densities than reported here. Based on the data presented in the present manuscript, an amendment of cultured cable bacteria was needed to stimulate its prolific growth in the rice pots. These are exciting findings, with potential direct practical applications.

In general the manuscript is very well written and the main conclusions are clearly presented and well substantiated. The key data, a decrease in methane emissions from rice pots with an inoculation of freshwater cable bacteria, are clear and compelling. Likewise, the authors present data showing strongly elevated sulfate concentrations in the zone of cable bacteria growth, and below, in the inoculated treatment over control. I find these data to be clear and persuasive. As a first study demonstrating beneficial environmental effects of cable bacteria amendments to soil of an important agricultural crop, the results are both novel and likely to be of high interest to a diverse interdisciplinary readership.

I concede there are some additional data I would have like to have seen, such as radiotracer measurements of sulfate reduction and/or methanogenesis. The authors must rely on inference for one of their arguments about the mechanisms underlying the observed decrease in methane efflux. Specifically, the authors suggest that methanogenesis and fermentation may be decreased by acidity attributed to cable bacteria activity. However, methanogenesis (or production or turnover of fermentation products) are not reported in this study, and the pH decrease shown is somewhat smaller than the once referenced from the literature (i.e., the decrease in pH shown in Figure 2b is less than from 7.0 to 6.5). Overall though, I consider this only a minor deficit, and I do not think this criticism outweighs the value of the overall results presented.

A few additional questions:

At what period during the 11 week incubation were the ~ 2 day methane flux measurements made? Forgive me if I missed this.

Did the cable bacteria inoculum affect the rice plant growth or appearance? It would be valuable to present the *Oryza* biomass (below ground and/or above ground), or at least to be assured that growth occurred throughout the experiment in both treatments and controls.

Can you report the approximate density (or total quantity added) of enrichment culture of *Ca. Electronema* sp. added to the treatment pots?

I'm curious about what might be controlling cable bacteria growth in these soils. Cable bacteria were enumerated at the end of 11 weeks of incubation, which indicates remarkable persistence. If they are limited by sulfide supply, then does that imply they are ultimately limited by the upstream rates of substrate supply to sulfate reducing bacteria (i.e., hydrolysis, fermentation?). I recognize this is fodder for future work.

Minor text suggestion from abstract:

Line 37: "The drastic reduction of methane emissions in pots with cable bacteria were likely caused by the combined effects of electrogenic sulfide oxidation which led to a 5-fold increased sulfate inventory"
- Here, "combined" is meant to refer to both the sulfate accumulation and pH decrease associated with e-SOX. However, without this context, "combined" is somewhat confusing because the reader is expecting "the combined effects of electrogenic sulfide oxidation and [...]".

- Consider as an alternative: The drastic reduction of methane emissions in pots with cable bacteria were likely caused by the *multifarious* effects of electrogenic sulfide oxidation which led to *an increase in acidity* and 5-fold increased sulfate inventory.

I'm signing this review, per request of editorial board.
Respectfully yours, Sairah Malkin

Reviewer #3 (Remarks to the Author):

This study shows that cable bacteria (Desulfobulbaceae) can mitigate methane emissions from a rice soil system. This is an important finding with implications for potentially mitigating methane emissions from rice field soils. However, the relevance of this process in natural rice field soil is not made clear. Presumably cable bacteria are present in rice field soils, which is why the authors constructed a soil lacking cable bacteria using autoclaved wetland soil and manure as a source of microbes lacking cable bacteria. This begs the question as to the extent to which this process occurs in natural rice field soils and thus the potential for it to be enhanced. The implication and impression when reading the manuscript is that the authors have uncovered a strategy to reduce the global warming impact of rice agriculture via inoculating rice fields with cable bacteria, but in fact this has not been investigated. At a minimum the authors should make this clear. Preferably they could report the density of cable bacteria in a natural rice field soil and estimate the extent to which this process occurs already in rice agriculture. In addition, I don't see why the authors have not performed this experiment with rice field soil to measure the potential of stimulating sulfur cycling by using *Electronema* as an inoculant.

A density of 250 m per cm³ observed in the top 1 cm of the soil seems like an incredibly high density. What was the average length of a filament / how many filaments does this correspond to? Worth mentioning if space permits that their (re)distribution in the top centimeters indicates that they are active as they evidently repositioned themselves between gradients of reductant and oxidant.

Response to referees

We thank the three reviewers for their constructive comments on the manuscript “NCOMMS-19-41942-T”. Our responses to the individual comments are highlighted in blue and the changed sections of the manuscript are included below the respective comment (shown in italic). Changes to fit the format guidelines and additional changes are highlighted in green. References to line numbers refer to the line numbering in the revised version of the manuscript.

Response to Reviewer #1:

In this manuscript, the authors show that inoculation of rice plant-vegetated soil pots with filamentous cable bacteria reduced methane emissions by 92%. They showed that sulfate concentrations increased significantly in pots amended with cable bacteria. This sulfate could then serve as an electron acceptor for sulfate reducers that were able to outcompete methanogens. Nice story. I enjoyed reading this paper. I just have a few minor comments but otherwise I think that this is an excellent paper.

Reply #1: We thank the reviewer for the positive feedback.

Line 34. 11-week incubation period, cable....

Reply #2: The abstract has been shortened to fit the formatting guidelines and this sentence has been deleted.

Lines 63-66. This sentence is awkward, please reword

Reply #3: We agree with the reviewer and the sentence has been rephrased:

“However, the externally amended sulfate can be quickly converted to sulfide, which tend to accumulate in the soil, implying that prolonged effects on CH₄ emissions can only be achieved by re-application of low amounts of sulfate^{16,17}.” [now lines 66-68].

Introducing the build-up of toxic sulfide levels has been omitted from the introduction and abstract as it should not be the focus of our story but is pointed out in the discussion.

Line 66. We hypothesized that cable bacteria could reduce methane emissions.....

Reply #4: The entire sentence has been changed to *“Our results indicate that cable bacteria reduce CH₄ emissions from rice-vegetated soils by recycling sulfate via e-SOX.”* to fit the formatting guidelines [now lines 73-74].

Line 71. After 11 weeks of incubation...

Reply #5: Done [now line 71].

Line 73. bacterial cells

Reply #6: Done [now line 81].

Line 100. What do you mean by domain? Are you referring to this specific region (upper 4 cm)? Please clarify.

Reply #7: *“Domain”* has been replaced by *“the first 4 cm of the soil”* [now line 97].

Line 106. Did it penetrate deeper? You shouldn't have to speculate. Why didn't you just measure sulfate concentrations below 4 cm?

Line 106-107. I'm not sure what you are getting at here. Please rewrite this section. Are you just trying to say that sulfate reducers are only present in the upper layers of the pots (where sulfate was found in the cable-free pots)???

Reply #8: This section has been rephrased for clarification:

“The experimental design did not allow to retrieve soil samples from the lower 3 cm of the pots, but the fact that the sulfate concentrations in the 3-4 cm depth section of the pots with cable bacteria were about twice as high as the concentrations in the top centimeter of pots without cable bacteria and sulfate yet penetrated 4 cm in the latter do imply, that sulfate penetrated to the bottom of the cable bacteria pots, assuming similar potential sulfate reduction rates.” [now lines 99-104].

Lines 136-137. But you didn't see drops in pH until after 4 cm of depth (Figure 2b). You didn't show data below 4 cm. What were sulfate concentrations like at depths where low pH would influence methanogenesis? Just curious.

Reply #9: We agree with the reviewer that the cable bacteria impact on pH was not clearly stated. Therefore we added the following sentences:

“Microprofiles of pH were only taken down to 7.2 mm depth to minimize the risk of sensor collision with the roots which would break the sensor. However, it has been shown previously that e-SOX produces acidity even down to 4 cm¹².” [now lines 133-136].

Line 152. Provide a reference showing that sulfate might improve plant performance by serving as a nutrient.

Reply #10: Done [now line 146].

Line 168. mixed with the autoclaved soil to provide an inoculum consisting of a complex microbial community

Reply #11: The sentence has been changed accordingly but moved from the Methods to the Discussion section [now lines 129-133].

Line 170. Please be clearer about the number of tablespoons that were added. Do you mean three tablespoons were added? Did you add exactly the same amount of cable bacteria to each amended pot?

Reply #12: We have now detailed the exact inoculation procedure:

*“The soil was then inoculated with cable bacteria by adding cable bacteria enrichment culture of *Ca. Electronema* sp. GS⁹ to the soil-cow dung mixture (≈1500:1, v/v). After carefully homogenizing the soil and filling into growth pots (depth:7 cm; diameter: 12 cm), the first 2 cm of the soil in each pot were further inoculated with the cable bacteria enrichment culture (≈225:1, v/v) and carefully homogenized.”* [now lines 163-167].

Reviewer #2 (Remarks to the Author):

The manuscript, “Cable bacteria reduce methane emissions from rice-vegetated soils” details an experiment in which rice seedlings were planted in pots with a mix of wetland soil and cow dung (as microbial inoculum) with or without an inoculum of cultured freshwater

filamentous cable bacteria (Ca. *Electronema* sp. GS). With the addition of the cable bacteria, the authors show sulfate was recycled, porewater pH was lowered, and methane emissions were mitigated to a remarkable degree. The authors suggest that the decrease in methane emissions were due to the combined effects of sulfate regeneration and acidity generation, as a result of cable bacteria activity near the sediment surface. Recently, Scholz et al. 2019 (FEMS) reported observing cable bacteria in the rhizosphere of several plants, including *Oryza sativa* (rice), though at far lower densities than reported here. Based on the data presented in the present manuscript, an amendment of cultured cable bacteria was needed to stimulate its prolific growth in the rice pots. These are exciting findings, with potential direct practical applications. In general the manuscript is very well written and the main conclusions are clearly presented and well substantiated. The key data, a decrease in methane emissions from rice pots with an inoculation of freshwater cable bacteria, are clear and compelling. Likewise, the authors present data showing strongly elevated sulfate concentrations in the zone of cable bacteria growth, and below, in the inoculated treatment over control. I find these data to be clear and persuasive. As a first study demonstrating beneficial environmental effects of cable bacteria amendments to soil of an important agricultural crop, the results are both novel and likely to be of high interest to a diverse interdisciplinary readership.

Reply #13: We thank Sairah Malkin for this positive feedback.

I concede there are some additional data I would have like to have seen, such as radiotracer measurements of sulfate reduction and/or methanogenesis. The authors must rely on inference for one of their arguments about the mechanisms underlying the observed decrease in methane efflux. Specifically, the authors suggest that methanogenesis and fermentation may be decreased by acidity attributed to cable bacteria activity. However, methanogenesis (or production or turnover of fermentation products) are not reported in this study, and the pH decrease shown is somewhat smaller than the once referenced from the literature (i.e., the decrease in pH shown in Figure 2b is less than from 7.0 to 6.5). Overall though, I consider this only a minor deficit, and I do not think this criticism outweighs the value of the overall results presented.

Reply #14: We agree that our mechanistic explanation is based on the cited references, namely Sandfeld et al. 2020, showing enhanced sulfate reduction rates induced by cable

bacteria activity and Wang et al. 1993, showing the correlation between the reduction of methane emissions and reduced pH in rice-vegetated soils. Measuring sulfate reduction rates with radiotracer in a heterogenous system with local sulfate depletions, like a vegetated soil, we consider a significant challenge for future studies. Moreover, we agree with Sairah Malkin that the reported reduction of 50% upon a pH decrease from 7.0 to 6.5 from the reference Wang et al. 1993 does not compare with the cable bacteria induced pH shift which we report, i.e. pH shift from 6.88 to 6.64 (0.24 pH units). We therefore changed the manuscript and now state different values from the reference Wang et al. 1993: *“For example, a pH shift from pH 6.9 to 6.8 has been reported to decrease CH₄ production in flooded rice soil by 26%²³.”* [now lines 137-139].

A few additional questions:

At what period during the 11 week incubation were the ~ 2 day methane flux measurements made? Forgive me if I missed this.

Reply #15: We acknowledge that it was not clearly stated. We have added the following sentences to the manuscript:

“After 11 weeks of incubation under water saturation, sulfate concentrations, methane emissions,...” [now lines 71-73].

“After the 11-week incubation period,...” [now line 176].

“Following the microelectrode measurements,...” [now line 184].

Furthermore, we believe that Supplementary Fig. 3 helps to understand the experimental procedure.

Did the cable bacteria inoculum affect the rice plant growth or appearance? It would be valuable to present the *Oryza* biomass (below ground and/or above ground), or at least to be assured that growth occurred throughout the experiment in both treatments and controls.

Reply #16: We agree and the following sentences have been added to the manuscript:

“..., no difference of the above-ground biomass of the rice plants between the treatments was observed (P=0.64, n=8, unpaired two-tailed t-test, see Supplementary Fig. 2),...” [now lines 77-79].

“After sampling, the above-ground biomass from each rice plant was collected and dried at 80 °C for 44 h to determine the dry weight.” [now lines 173-174].

“The dry weight of the above-ground biomass,...” [now line 204].

“The above-ground biomass of single plants grown in soil with cable bacteria was 3.4 ± 0.3 g dry weight and in soil without cable bacteria 3.6 ± 0.3 g dry weight (mean \pm s.e.m., n=4).”
(see figure legend of Supplementary Figure 2).

Can you report the approximate density (or total quantity added) of enrichment culture of *Ca. Electronema* sp. added to the treatment pots?

Reply #17: This comment coincides with the comment of the first reviewer and the manuscript has been changed accordingly (see reply#12). Furthermore, we would like to point out that the inoculation procedure, i.e. spreading cable bacteria enrichment culture in the upper 2 cm cannot explain the observed higher cable bacteria densities in the upper cm of the soils after 11 weeks of incubation. If cable bacteria originated solely from the inoculation without any further growth, the cable bacteria density of the inoculum would have needed to be at least 90000 m cm^{-3} . Such a high density has never been reported and is hardly imageable. In other words, cable bacteria have grown and persisted throughout the incubation period.

I'm curious about what might be controlling cable bacteria growth in these soils. Cable bacteria were enumerated at the end of 11 weeks of incubation, which indicates remarkable persistence. If they are limited by sulfide supply, then does that imply they are ultimately limited by the upstream rates of substrate supply to sulfate reducing bacteria (i.e., hydrolysis, fermentation?). I recognize this is fodder for future work.

Reply #18: Indeed, the persistence in our laboratory incubation could be explained by the intense sulfur cycling fueled by root exudation, whereas cable bacteria in a closed system might “starve” after such a long incubation. We have added the following sentence to the manuscript to point out that roots provide a continuous source of organic carbon:

“These substrates were supplied from fermentations processes, which were fueled by organic carbon from root exudations and sloughed root material¹⁹” [now lines 127-128].

Minor text suggestion from abstract:

Line 37: “The drastic reduction of methane emissions in pots with cable bacteria were likely caused by the combined effects of electrogenic sulfide oxidation which led to a 5-fold increased sulfate inventory”

- Here, “combined” is meant to refer to both the sulfate accumulation and pH decrease associated with e-SOX. However, without this context, “combined” is somewhat confusing because the reader is expecting “the combined effects of electrogenic sulfide oxidation and [...]”.

- Consider as an alternative: The drastic reduction of methane emissions in pots with cable bacteria were likely caused by the *multifarious* effects of electrogenic sulfide oxidation which led to *an increase in acidity* and 5-fold increased sulfate inventory.

I’m signing this review, per request of editorial board.

Respectfully yours, Sairah Malkin

Reply #19: This suggestion has been implemented in the main text [now lines 139-140]. Due to shortening of the abstract this sentence has been omitted from the abstract.

Reviewer #3 (Remarks to the Author):

This study shows that cable bacteria (Desulfobulbaceae) can mitigate methane emissions from a rice soil system. This is an important finding with implications for potentially mitigating methane emissions from rice field soils. However, the relevance of this process in natural rice field soil is not made clear. Presumably cable bacteria are present in rice field soils, which is why the authors constructed a soil lacking cable bacteria using autoclaved wetland soil and manure as a source of microbes lacking cable bacteria. This begs the question as to the extent to which this process occurs in natural rice field soils and thus the potential for it to be enhanced. The implication and impression when reading the manuscript is that the authors have uncovered a strategy to reduce the global warming impact of rice agriculture via inoculating rice fields with cable bacteria, but in fact this has not been investigated. At a minimum the authors should make this clear. Preferably they could report the density of cable bacteria in a natural rice field soil and estimate the extent to which this process occurs already in rice agriculture.

Reply #20: Indeed, the occurrence of cable bacteria in rice fields has not been investigated yet. We modified the last paragraph of our manuscript to point out that this is the first study showing the correlation between cable bacteria and methane emissions, which opens up for many more questions including the in-situ activity of cable bacteria in rice fields.

“Our finding arises the questions to what extent cable bacteria grow in wetland rice fields and whether their presence can be promoted by...” [now lines 152-156].

In this context, we have also omitted the last 5 lines at the end of the discussion section (*“In conclusion, the large potential of cable bacteria to colonize freshwater systems, combined with the electrogenic imprints, indicate that cable bacteria could be key players in future mitigation strategies of CH₄ emissions from water-saturated soils. The effects of cable bacteria on CH₄ emissions are not limited to rice fields but extend to all other freshwater and marine wetlands, highlighting the crucial role of cable bacteria in protecting our climate.”*, previous lines 156-161), as it appeared repetitive and we do not show that the effects of cable bacteria on reducing methane emissions hold true for other freshwater and marine wetlands.

In addition, I don't see why the authors have not performed this experiment with rice field soil to measure the potential of stimulating sulfur cycling by using *Electronema* as an inoculant.

Reply #21: The wetland soil was autoclaved before inoculation with cow dung and cable bacteria, therefore no contribution from native microbial communities were to be expected, no matter if it was rice soil or wetland soil. We agree that the rationale of mixing cow dung into the wetland soil was not well enough highlighted and therefore we have added the following section to the discussion:

“Moreover, we chose autoclaved wetland soil as universal matrix to test our hypothesis. This wetland soils was supplemented with cow to provide an inoculum consisting of a complex microbial community including fermenters, methanogens and sulfate-reducing bacteria²⁰ but no cable bacteria, which also increased the initial organic carbon pool in the incubation pots at the beginning of our experiment.” [now lines 128-133].

A density of 250 m per cm³ observed in the top 1 cm of the soil seems like an incredibly high density.

Reply #21: This density is not exceptional and the following sentence has been added for clarification:

“Cable bacteria successfully established after one-time inoculation and the filament density was well within the typical range of cable bacteria abundance^{6,26”} [now lines 147-148].

What was the average length of a filament / how many filaments does this correspond to?

Reply #22: The cable bacteria density was determined by Fluorescence in situ hybridization and the processing of the samples may have sheared the long cable bacteria filaments, so that the actual lengths of individual filaments cannot be reported.

Worth mentioning if space permits that their (re)distribution in the top centimeters indicates that they are active as they evidently repositioned themselves between gradients of reductant and oxidant.

Reply #23: We agree that growth of cable bacteria evidently correlates to cable bacteria activity. However, after the growth phase cable bacteria might become substrate limited (cf. Reply#18) and their activity might cease while their filaments persist. In our study we showed that cable bacteria were still active after 11 weeks of incubation by pH microprofiling, because the cable bacteria-induced drop of pH would quickly vanish after the cessation of cable bacteria activity.

Additional changes in the manuscript to fit the format guidelines:

- The abstract has been shortened, rephrased and references were removed.
- The “Extended Figures” have been changed to “Supplementary Figures” in the main text and a “Supplementary Information” file has been created.
- Result section with subheadings and discussion section have been added, which implied relocation of specific sections to the discussion: *“which was uniformly distributed in the upper 4 cm in pots with cable bacteria, suggesting that sulfate*

reduction was balanced by sulfur re-oxidation via e-SOX and eventually that ionic migration adds to the transport of sulfate¹².” [now lines 122-125] and “Cable bacteria were also found on roots which is congruent with previous studies that report the enrichment of cable bacteria on oxygen-releasing plant roots^{27,28}. Indeed, rice roots can release oxygen²⁹ providing the electron acceptor for cable bacteria. Thus, wetland rice fields might constitute an ideal habitat for cable bacteria” [now lines 148-152].

- Numbering of subheadings from Methods section has been removed.
- The last sentence of the Introduction has been changed to “*Our results indicate that cable bacteria reduce CH₄ emissions from rice-vegetated soils by recycling sulfate via e-SOX.*” [now lines 73-74].
- Figures have been removed from manuscript file and uploaded as separate files in the submission system. Figure Legends have been moved to the end of the main text file.
- Data availability statement has been added.

Additional changes:

- Sandfeld is now cited as published reference.
- The following values have been replaced for correctness: “*41 ± 9 μmol m⁻² day⁻¹*” has been replaced by “*42 ± 9 μmol m⁻² day⁻¹*” [now line 116], “*92%*” has been replaced by “*93%*” [now lines 32, 117, 120], “*7 mm*” has been replaced by “*7.2 mm*” [now line 106, 134]. Fig. 3 has been redone. Note that these changes do not influence any other changes in the manuscript.
- Fig. 4 has been modified (cloud around methane was added and red lines enlarged).
- Words have been replaced to improve clarity and are highlighted in the manuscript file in green. E.g. “*at the soil surface*” has been deleted and replaced by “*in the upper soil layers*” [now line 85].
- Line thickness and colour in Fig. 2a have been adjusted.

REVIEWERS' COMMENTS:

Reviewer #1 (Remarks to the Author):

I feel that the authors have satisfied all of the reviewer's requests.

Reviewer #2 (Remarks to the Author):

The authors responses were thorough and thoughtful and fully addressed my (minor) concerns. I have no further comments.

Sairah

Reviewer #3 (Remarks to the Author):

I reviewed the first submission (Reviewer 3). I've read the response and the revised manuscript and am satisfied with the answers and the revisions. I feel that the study makes a highly valuable contribution to understanding of wetland microbial ecology and nutrient cycling and I have no further comments.

Response to referees

We thank the three reviewers for reviewing our manuscript.

REVIEWERS' COMMENTS:

Reviewer #1 (Remarks to the Author):

I feel that the authors have satisfied all of the reviewer's requests.

Reviewer #2 (Remarks to the Author):

The authors responses were thorough and thoughtful and fully addressed my (minor) concerns. I have no further comments.

Sairah

Reviewer #3 (Remarks to the Author):

I reviewed the first submission (Reviewer 3). I've read the response and the revised manuscript and am satisfied with the answers and the revisions. I feel that the study makes a highly valuable contribution to understanding of wetland microbial ecology and nutrient cycling and I have no further comments.